



# Sub-seasonal snowline dynamics of glaciers in Central Asia from multi-sensor satellite observations, 2000-2023

Dilara Kim[1,2], Enrico Mattea[1], Mattia Callegari[2], Tomas Saks[1], Ruslan Kenzhebayev[3], Erlan Azisov[3], Tobias Ullmann[4], Martin Hoelzle[1], and Martina Barandun[1,2]

[1]Department of Geosciences, University of Fribourg, Fribourg, Switzerland.
[2]Institute of Earth Observation, EURAC, 39100 Bolzano, Italy.
[3]Central-Asian Institute for Applied Geosciences (CAIAG), Bishkek 720027, Kyrgyzstan.
[4]Department of Remote Sensing, Institute of Geography and Geology, University of Würzburg, Würzburg, Germany.

**Correspondence:** Dilara Kim (dilara.kim@unifr.ch)

**Abstract.**

Sub-seasonal glacier dynamics strongly influence the timing and magnitude of meltwater supply, a vital component of summer runoff in dry Central Asia region. Understanding of snowline evolution during the melt season is therefore essential for predicting seasonal water availability and glacier response to climate change. We present a novel method to infer 24-years of sub-seasonal snowline dynamic for four glaciers distributed throughout Pamir and Tien Shan mountain ranges using multi-sensor spaceborne observations. Our approach combines medium-resolution optical MODIS with high-resolution Sentinel-2 optical and Sentinel-1 radar imagery to produce close-to-daily estimates of the glacier Snow-Covered Area Fraction (SCAF - the ratio between snow covered area above the snowline and the total glacier area) throughout the melt season from 2000 to present. The method was validated against manually delineated Landsat snowlines, achieving RMSE values below 20% for most sites. The resulting time series reveal substantial interannual and regional snowline variability with e.g. June SCAF ranging between 60-100%. Recent warm years, show earlier exposure of bare ice and shifts in the melt season's end by as much as a month later in September. Accelerating snow depletion rates were found for all four glaciers, starting in 2000 and 2009 and reaching up to -1.25%/day. Linking these dynamics to the annually measured and daily modelled mass balance data highlights that similar annual mass balance values can have large differences in sub-seasonal snow depletion and thus meltwater contribution, with implications for water availability during the critical dry-season months. Our findings demonstrate the potential of long-term, high-temporal-resolution snowline monitoring to improve understanding of glacier-climate interactions and to better constrain seasonal runoff forecasts in Central Asia's water-scarce river basins.

## 1 Introduction

Glaciers are recognized as essential climate variable (World Meteorological Organization (WMO), 2022). The consequences of glacier mass changes are drastic, as the glacier meltwater impacts the surrounding ecosystems (Milner et al., 2017; Huss et al., 2017), and regional food and water security (Huss and Hock, 2018). Central Asian mountain ranges, Pamir and Tien Shan encompass over 25,000 glaciers. The meltwater from these glaciers shapes the annual hydrological regime of the region,




making significant contribution during dry summer period. Therefore, the accurate estimation of the glacier melt and the sub-seasonal change is crucial (Barandun et al., 2020).

During the Soviet times, the reference glaciers (e.g. the Abramov glacier) in the region were comprehensively monitored throughout the entire season. The abrupt interruption of observations happened after the dissolution of the Soviet State; how-ever, since 2010 the systematic efforts to re-establish the monitoring network gradually led to continuation of long-term annual mass balance measurements (Schöne et al., 2013; Hoelzle et al., 2017). Despite bringing invaluable reference information on the glacier state, the in-situ monitoring remains labor intensive and limited spatially (Zemp et al., 2015). Remotely observable proxies for glacier response to climate change such as variation in the extent of the accumulation area ratio (AAR) (Dyurg-erov et al., 2009; Bahr et al., 2009), alteration in median glacier elevation (Sakai et al., 2015; Sakai and Fujita, 2017) or the area-altitude distribution (Tangborn, 1999) have been widely utilized (Rabatel et al., 2017). One of the most common methods is equilibrium-line altitude (ELA)/AAR concept. The ELA is defined as the mean altitude on a glacier where accumulations equals the ablation, i.e. the point with the zero annual surface mass balance (Cogley et al., 2011). High correlation between the ELA or AAR and the glacier-wide surface mass balance (SMB) was observed for a wide range of glaciers (Østrem, 1975; Braithwaite, 1984; Kuhn, 1990). Usually this relation is linear (Dyurgerov, 2010). However the ELA and AAR cannot be ob-served directly but approximated through observations of the end-of-summer snowline on a glacier on remote sensing images (Kulkarni, 1992; Rabatel et al., 2005). In order to create a statistically significant relationship a long-time series of annual mass balance and end-of-summer snowline is thus needed (Dyurgerov, 1996). Dyurgerov (1996) included the transient snowline and the transient surface mass balance to have a significant amount of observations to build the relationship. Previously, the sea-sonal snowline rate of change was calculated by (Pelto, 2011) between the available Landsat observations for the Taku Glacier, Alaska. The author tied the snowline migration rate to changes in mass balance gradient and concluded on the potential for an efficient mass balance assessment from high temporally resolved snowline time series for many years and on many glaciers.

The end-of-season snowline marks the transition between snow and bare ice surfaces and thus approximate limit between the accumulation and ablation area. The studies outlined above often refer to the snow-covered area on a glacier (SCA) instead of the snowline altitude or transient snowline (SLA or TSL). This is done to avoid the influence of glacier location and geometry when establishing the statistical relationship with the surface mass balance. In our work, we use the Snow-Covered Area Fraction (SCAF), which is defined as the area of the glacier surface that is snow-covered in relation to the total glacier area. Different methods existed to distinguish snowlines on the glaciers. Before the age of satellites, the snowline altitude (SLA) measurements were conducted in the field using various instruments such as theodolite, aneroid and inclinometer (Braithwaite, 2015; Kalesnik, 1963). These methods are however labour-intensive, and unlike ablation and accumulation measurements, not commonly conducted. The regional studies were possible with aerial surveys and the co-registration of aerial photographs with large scale topographic maps (Østrem, 1973; Konovalov, 1962).

The second half of the 20th century marked the breakthrough in snowline monitoring, as snow and ice have a distinctive spectral signatures in the near infrared (NIR) and visible part of electromagnetic spectrum, making them suitable for mapping from optical satellite imagery during the melt season (Østrem, 1975; Hall et al., 1989). Snow in accumulation area typically has a high reflectance, while the bare ice reflectance in ablation zone is much lower. Therefore, single-band and band-ratios





have been long used to determine the border between snow-covered and ice surfaces on a glacier (Williams et al., 1991). One of the classical band-ratio approach to determine the presence of snow cover is the normalized difference snow index (NDSI), which is a normalized difference between green and shortwave-infrared (SWIR) bands. Multiple studies used NDSI to determine SLA based on different thresholds (Dozier, 1989; Girona-Mata et al., 2019; Hall et al., 1995; Kaur et al., 2010). The other techniques utilize band-ratios, such as Red/SWIR (Winsvold et al., 2016), NIR/SWIR (Hall et al., 1987), $NIR^2$/SWIR (Davaze et al., 2020). Rastner et al. (2019), however, argued band ratios are more suitable to map snow and ice together as one class, as they have similar spectral response. Therefore, Rastner et al. (2019) proposed to use the single-band reflectance values (e.g NIR), as in (Rabatel et al., 2012; Racoviteanu et al., 2019). An alternative method is the use of available albedo products (Brun et al., 2015; Naegeli et al., 2019). More recent methods applied machine learning to extract SLA (Prieur et al., 2022; Zeller et al., 2025; Aberle et al., 2025), but as it relies on manually acquired training dataset, the transferability of such methods is limited.

Majority of above mentioned studies used the Landsat mission for the classification of snow and ice due to the combination of spatial resolution (30 m) and longer time series. Both manual (e.g. Larocca et al. (2024)) and automated (e.g. Racoviteanu et al. (2019); Loibl et al. (2025)) approaches are described. However, the limited revisit time of 16 days and the influence of cloud cover significantly reduce the number of useful scenes, so that the transient snowline evolution during the ablation period as well as the end-of-season snowline may not be captured correctly. Moderate Resolution Imaging Spectroradiometer (MODIS) instrument offers 1-day temporal revisit, but was used only to calculate regional snowlines due to a coarse spatial resolution (Shea et al., 2013; Spiess et al., 2015). Brun et al. (2015) used mean MODIS albedo products on individual glaciers in Himalaya as a proxy for the snowline and eventually the mass balance.Aberle et al. (2025) derived SLA from Landsat, PlanetScope and Sentinel-2 between 2013 and 2022 for glaciers in Alaska. The PlanetScope and Sentinel-2 offered the superior spatial and temporal resolution, but a shorter time-series than Landsat and MODIS. In contrast to the weather-dependent optical imagery, several studies showed the utility of radar data in C-band to map the snowline (Adam, 1997; Rees et al., 1995). As the snow starts to melt, it exhibits a much lower backscattering than bare ice, which allows to map snowline (Rott and Mätzler, 1987). The dry snow is however transparent to radar signal, so that the fresh snowfalls are not detected. Winsvold et al. (2018) demonstrated the possibility to track the snowline on the glacier from Sentinel-1. Other methods are using the polarimetric SAR to detect the snowline (Callegari et al., 2016)

While the latest generation of satellites improved in spatial and temporal resolutions, the long-term glacier snowline observations still rely on Landsat. In this work we propose a novel method $glacierSCAF_{MODIS}$ to extract the snowline at the glacier scale based on optical and radar sensors at various spatial and temporal resolutions. We harness the advantages of daily MODIS acquisitions and modern high resolution images of Sentinel-1 and Sentinel-2 (Sect. 3.1). The snowline time-series for glaciers in Central Asia is closely examined (Sect. 4.1.1) and validated (Sect. 4.1.2). We provide the insight on the sub-seasonal snowline dynamics in Section 4.2.1. In Section 3.2 we describe the metrics we extracted from the $glacierSCAF_{MODIS}$ results and further inspect them in Section 4.2.2. We show the potential to utilize obtained transient snowlines for surface mass balance model calibration (described in Sect. **??**) in order to improve sub-seasonal mass balances and glacier melt water contribution (Sect. 4.3.2). The link between snowline dynamics, mass balance and implication for changes in meltwater contribution is



discussed in Section 5.1. We also assess the technical challenges in Section 5.2 and link our results to the mass balance of the glacier (Sect. 5.3).

## 2  Study Sites and Data

### 2.1  Study Sites

This study focuses on four glaciers located in the different regions of the Pamir and Tien Shan mountains in Central Asia. The overall dry and cold mountain ranges are characterized by contrasting glacier mass balance responses (Fig. 1) (Barandun et al., 2021; Barandun and Pohl, 2023; Hugonnet et al., 2021). Glaciers in Northern/Western Tien Shan and Pamir Alay (Abramov, Golubin) are strongly influenced by westerlies and are receiving most of its precipitation in late winter and spring. (Pohl et al., 2015; Aizen et al., 1995, 2009). Conversely, due to the orographic barrier effect most of the glaciers of the Central Tien Shan (Glacier No. 354) are shielded and receive most of the precipitation in summer, when Siberian anticyclones bring cold and moist air masses (Schiemann et al., 2008). Orographic shielding creates increasingly arid conditions to the glaciers in central Pamir (Zulmart) all year round. Due to the different topo-climatic effects, the four glaciers studied (Table 1), represent different topo-climatic settings and responses to changes in atmospheric conditions (Barandun et al., 2021)

**Table 1.** Overview of the studied sties. Reference years are the periods when in situ mass balance measurements are conducted.

| Glacier | Abramov | Golubin | Glacier No. 354 | Zulmart |
|---|---|---|---|---|
| **Location** | 39.6200, 71.5600 | 42.4600, 74.4950 | 41.7990, 78.1506 | 38.8631, 72.9989 |
| **Area, km$^2$** | 22.55 | 5.454 | 6.4 | 3.66 |
| **Region** | Pamir-Alay | Northern/Western Tien Shan | Central Tien Shan | Pamir |
| **Length (km)** | 7.7 (2021) | 4.7 (2021) | 4.4 (2021) | 3.9 (2018) |
| **Elevation (m, a.s.l.)** | 3659-4918 | 3400-3400 | 3841–4669 | 4600-5470 |
| **Reference years** | 1967-1998, 2011- | 1958-1994, 2011- | 2010- | 2018- |

### 2.1.1  Abramov

Abramov glacier is the largest of four study glacier with approx. 21 km$^2$ as of 2024 (**?**). It is located in Pamir Alay (north-western Pamir) in the south of Kyrgyzstan within the basin of the Vakhsh river (Koksu river), flowing into the Amu Darya river. Atmospheric, cryospheric and hydrological monitoring of the glacier started in 1967 and lasted until forced interruption in 1999 (Suslov et al., 1980; **?**; Pertziger, 1996). Close to the glacier tongue at 3837 m a.s.l. the mean air temperature (1968-1998) was $-4.1°C$ and annual precipitation sums with a peak in spring months for the same period are 750 mm (Pertziger, 1996). Direct surface mass balance observations were re-established in 2011 (Hoelzle et al., 2017). The re-established monitoring contains annual glaciological measurements (Barandun et al., 2015), snowline monitoring using terrestrial cameras and remote sensing (Barandun et al., 2018), meteorological monitoring through an automatic weather station (AWS, Schöne et al. (2013)) and was complemented with hydrological monitoring since 2019 and flow dynamics observations since 2022. Several studies



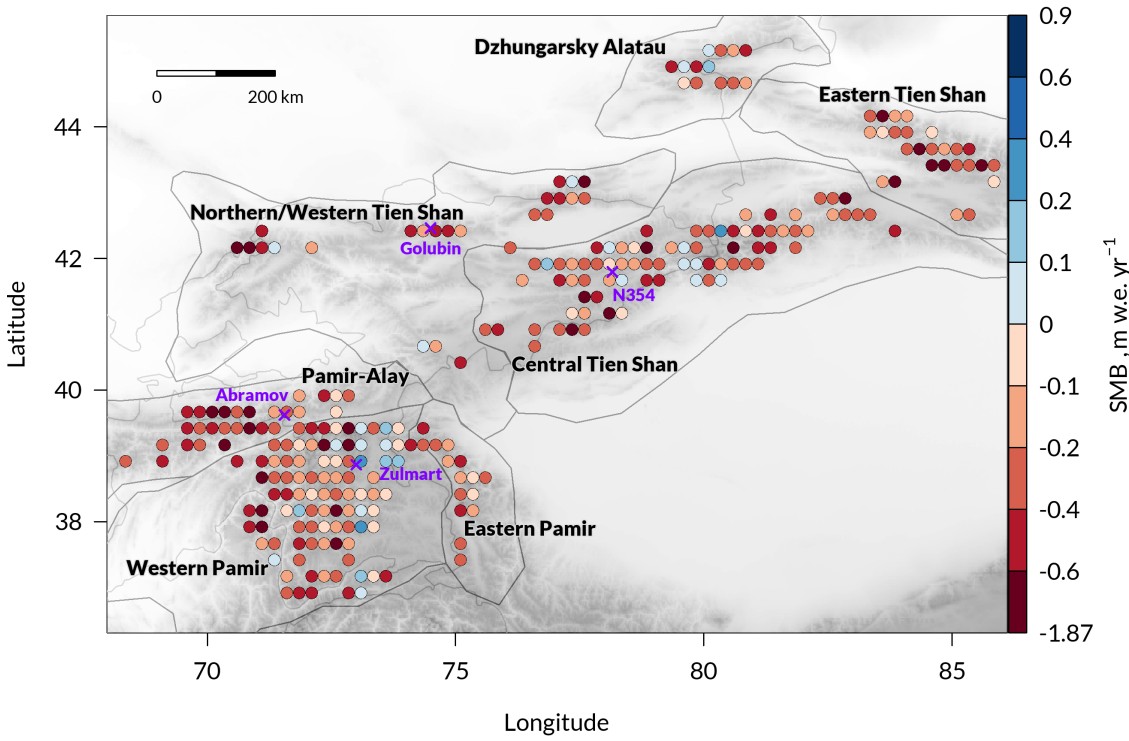

**Figure 1.** Overview map of the study region. The pie charts represent area-weighted mass balance estimated from (Barandun et al., 2021) between 2000-2018. Four study glaciers are marked in purple. The figure is adapted from Barandun et al. (2021)

reconstructed glacier mass balance for the past decades (**?**Kronenberg et al., 2021, 2022; Denzinger et al., 2021). The average annual mass balance was $-0.452\,\mathrm{m\,w.e.\,yr^{-1}}$ between 1968-1998 and $-0.365\,\mathrm{m\,w.e.\,yr^{-1}}$ between 2012-2023 (WGMS, 2024)

### 2.1.2 Golubin

Golubin glacier is located in the Ala–Archa catchment at the Kyrgyz Ala-too range in the northern Tian Shan in Kyrgyzstan.
The Ala–Archa catchment is the part of Chu river basin and main water resource for Bishkek, the capital city of Kyrgyzstan. Active glacier monitoring was conducted between the years 1958-1994 (Aizen, 1988). Surface mass balance measurements were resumed in 2010 (Hoelzle et al., 2017). An AWS was installed at an elevation of 3300 m a.s.l., situated 500 m from the glacier in 2013 Schöne et al. (2013). Another meteorological station Alplager, is situated 10 km down the valley from the glacier and at an elevation of 2145 m a.s.l. It recorded average mean temperature 3.23°C (1980-2019). The region receives
annual precipitation sums of approximately 700 mm with the major portion falling through April to June (Aizen et al., 2006). The average annual mass balance is $-0.303\,\mathrm{m\,w.e.\,yr^{-1}}$ between 1969-1994 and $-0.389\,\mathrm{m\,w.e.\,yr^{-1}}$ between 2011-2023 (WGMS, 2024)



### 2.1.3 Glacier No. 354

Glacier No. 354 is located in Akshiirak range in the Central Tien Shan in Kyrgyzstan. The glacier meltwater feeds into the
Naryn river and Syr Darya. The accumulation zone consists of three tributaries. The AWS station is owned by Kumtor Gold
Mines at 3660 m and recorded average precipitation of 360 mm in 1997–2014 with a peak between May-August (Kutuzov and
Shahgedanova, 2009). The meteorological data are available starting from 1930 from the Tien Shan meteorological station near
the AWS (Kutuzov and Shahgedanova, 2009), but no historical glaciological records exist. Surface mass balance monitoring
was initiated in 2010. The glacier experiences frequent summer snowfalls, which cover the entire glacier for several days
slowing down the melt (Kronenberg et al., 2016). Also, Kronenberg et al. (2016) observed extensive superimposed ice zone
on the glacier and estimated internal accumulation at +0.04 m w.e. yr$^{-1}$. The average annual mass balance was −0.730 m w.e.
yr$^{-1}$ between 2011-2023 (WGMS, 2024)

### 2.1.4 Zulmart

Zulmart (East Zulmart or Glacier No.139) is located to the south west of the Karakul lake in Tajikistan. It feeds Sarygun river,
which later becomes Akdjilga river and flows into Karakul lake. The temperature regime of this part of Pamir is characterized
by cold winter and cool summers with high annual and daily air temperature amplitudes (Atlas et al., 1975). The region is
extremely dry, receiving around 108 mm of precipitation per year with maximum between May and June recorded at Karakul
station (Gidrometeoizdat, 1969). Air temperatures measured at the front of the glacier indicate annual mean temperatures
of −7°C from 2019 to 2024. The glacier's high elevation and cold and dry climate suggest a strong radiation dominated
regime. Sublimation might thus be an important contributor to mass loss and a substantial amount of accumulation might relate
to the refreezing of snow melt, hereinafter referred as superimposed ice. These processes remain poorly known in the region.
Continuous glacier monitoring has been established in 2018 (Barandun et al., 2020). The average annual mass balance between
2019-2023 is estimated to be -0.252 m w.e. yr$^{-1}$ (WGMS, 2024).

## 2.2 Satellite data

To compute SCAFs over glaciers we employed both optical and radar satellite imagery acquired through the melt season
between 1st of June and 30th of September of 2000-2023. All products are available at the Google Earth Engine cloud platform
(GEE) (Gorelick et al., 2017). The Appendix A1 provides the summary on the temporal coverage and frequency of revisit of the
satellite missions that were used for the SCAF estimation and for validation. . Glacier No. 354 has persistent cloud coverage,
so less scenes from optical satellites were available.

### 2.2.1 Optical data

#### *MODIS*

The MODIS instrument was launched onboard the Terra spacecraft on December 18, 1999, became operational in March
2000, and provides daily global coverage across 36 spectral bands. We used atmospherically corrected surface reflectance



product MOD09GA version 61 with the NIR band at wavelength of 841–876 nm and pixel size of 250 m. The information on the
cloud state is obtained from the state_1km bitwise band. In particular, we used bits 0–1 to mask the clouds at 1 km resolution,
which is based on MODIS cloud product MOD35. The great advantage of MODIS is its long observation records and high
temporal revisit available at the coarse spatial resolution. As the glaciers retreated over a 20 year period, the exposed rock
within the glacier outlines became more prominent. Both reflectances from rock and glacier ice are lower than the reflectance
of snow in NIR band.

*Sentinel-2*

The Sentinel-2 MultiSpectral Instrument (MSI) is carried by a constellation of two polar-orbiting satellites: Sentinel-2A and
Sentinel-2B. They capture radiation reflected from the earth surface in 13 bands, including the NIR band centered at 842 nm and
10 m resolution. As the Sentinel-2B was launched in 2017, the imagery in 2016 was collected solely by Sentinel-2A, resulting
in a revisit time of 5-15 days. From 2017, the frequency of observation increases to 2-5 days for all glaciers. We used top-of-
atmosphere surface reflectance product Sentinel-2 Level-1C. The clouds are masked with Sentinel-2 Cloud Probability product
developed by Sentinel Hub. The cloud mask is derived from s2cloudless algorithm developed by Sinergise's EO Research
team (https://github.com/sentinel-hub/sentinel2-cloud-detector). Sentinel-2 MSI provides the high spatial resolution suitable
to extract SCAF over glaciers; however, the acquisitions started only in 2016.

### 2.2.2    Radar

*Sentinel-1*

Sentinel-1 operates two near polar sun-synchronous orbit satellites at low Earth orbit with a 6-day revisit (as constellation).
Both satellites feature a synthetic radar aperture (SAR) instrument onboard at the central frequency of 5.407 GHz (C-band). The
collected imagery is weather-independent, which is an advantage for glaciers with persistent cloud-cover during the summer.
As the target period covers the entire archive, we used available Level-1 Ground Range Detected (GRD) without the phase
information. The Interferometric Wide swath (IW) mode is acquired in dual polarization VV and VH over Central Asia and
features the spatial resolution of 20 m at ground range and 22 m in azimuth direction re-sampled on a 10 x 10 m grid since
2016.

### 2.3    Auxiliary Data

### 2.3.1    Validation data

To validate our automatic mapping tool we used manually delineated snowlines from Landsat imagery from Barandun et al.
(2018, 2021) that span between 2000 and 2016 for Abramov, Golubin and Glacier No. 354. For more details on the data and
methods consult Barandun et al. (2015). We delineated snowlines manually for Zulmart for period between 2018-2023 on
Landsat images.



### 2.3.2 Glacier outline

We use glacier outlines (Table 2) provided by the glacier inventories of RGI (RGI Consortium, 2017) and GLIMS (GLIMS Consortium, 2005). We kept glacier outlines constant over time.

**Table 2.** Glacier outlines ID in RGI 6.0 and GLIMS databases

| Glacier | RGI-Id | GLIMS ID | Date |
|---|---|---|---|
| Abramov | RGI60-13.18096 | G071570E39610N | Jul 10 2002 |
| Golubin | RGI60-13.11609 | G074498E42454N | Aug 24 2000 |
| Glacier No. 354 | RGI60-13.07064 | G078164E41793N | Aug 25 2002 |
| Zulmart | RGI60-13.14451 | G072999E38863N | Sep 28 2001 |

### 2.3.3 DEM

The digital elevation model is used for topographic and snowline altitude (SLA) corrections. We selected NASADEM offered at the spatial resolution of 30 m. It is a reprocessed SRTM with increased accuracy incorporating data from ASTER GDEM, ICESat GLAS, and PRISM datasets NASA JPL (2020).

### 2.3.4 Surface mass balance data

Glacier mass balance observations on annual resolution are based on a reanalysis of the point measurements published in WGMS (2024). Direct observations are available on annual resolution for Abramov since the mass balance year 2011/12, for Golubin and Glacier No. 354 since 2010/11 and for Zulmart only since 2018/19 (Barandun et al., 2025a). The point observations have been extrapolated to the entire glacier area by using a model-based extrapolation tool *DMBSim* provided in Mattea (2025). Using such a physical-based approach allows for an automated extrapolation of in situ observations that takes into account terrain characteristics and snow redistribution processes such as avalanche depositions or wind redistribution. It furthermore can extrapolate the measured mass balance from a float system to the hydrological year. *DMBSim* thus helps to obtain a better representation of the surface mass balance, especially for unmeasured and inaccessible areas of each glacier and corrects for inconsistent measurement periods. The reanalyzed annual mass balances are obtained from joint and ongoing glacier monitoring efforts in Central Asia to repeatedly homogenizing current glacier observation time series of the region in a systematic way (Barandun et al., 2015; Kronenberg et al., 2016; Kenzhebaev et al., 2017; Azisov et al., 2022; Severskiy et al., 2024). Data for the measured period are published through the WGMS (2024), whereas the data for the hydrological year have so far not been published and were provided by the corresponding research institutes.

### 2.3.5 Daily mass balance time series

Modeled annual mass balances were provided from the homogenization efforts described in Sect. 2.3.4, that delivers as a byproduct daily time series of surface mass balances, melt and SLA/SCAF. To obtain daily time series a distributed accumu-





lation and enhanced temperature index melt model developed by Mattea (2025) was calibrated to annual in situ observations available for each glacier. The model was run with mean daily temperature and daily precipitation sums. Air temperature measurements are available from nearby AWS for all sites (Barandun et al., 2025a). Precipitation is measured at the Alplager meteorological station for Golubin (Azisov et al., 2022) and at the Tien Shan AWS for Glacier No. 354 (Kronenberg et al., 2016; Kenzhebaev et al., 2017). For Abramov and Zulmart glacier, daily total precipitation sums were obtained from downscaled ERA5 data (Hersbach et al., 2020) using TopoScale (Fiddes and Gruber, 2014). Site specific parameter choices are based on existing studies from literature (Glazirin' et al., 1993; Suslov et al., 1980; Aizen et al., 1995; Kronenberg et al., 2016; Azisov et al., 2022). A detailed description of the model and calibration setup is given in Mattea (2025).

## 3 Methods

We developed a novel approach to infer the close-to-daily SCAFs for the period before high-resolution satellites became available. The method combines medium-resolution MODIS time-series, high-resolution multispectral Sentinel-2 and cloud-independent Sentinel-1 SAR imagery. We pre-process MODIS imagery to obtain mean reflectance over the glacier (MODIS meanNIR, see Sect. 3.1) and extract SCAF from Sentinel-1 (SCAF S1, see Sect. 3.1) and Sentinel-2 imagery (SCAF S2, see Sect.3.1). The satellite scenes are processed by the cloud computing service of GEE. From the dates when both MODIS and either Sentinel-1 and Sentinel-2 observations are available we derive a statistical relation (see Sect. 3.1.3) to predict high resolution MODIS-based SCAF. The MODIS-derived SCAFs are validated with Landsat manual delineations. Figure 2 illustrates the main stages of the workflow. From the retrieved time-series we calculated the end-of-season SCAF and annual rate of change (see Sect.3.2). Finally, we used the SCAF produced by glacier mass balance model described in Section 2.3.5 and compared the modelled SCAF to MODIS-based SCAF.

### 3.1 $glacierSCAF_{MODIS}$ algorithm

#### MODIS meanNIR

As the spatial resolution of MODIS is too coarse for a direct pixel-wise snow cover classification on glacier-scale, we calculated mean reflectance value over the total glacier area instead. To include all relevant MODIS pixels, we created a buffer around each glacier. The buffer area is approximately 1.4 times original glacier area. All pixels covering less than 65% of the buffered area are excluded. To filter cloud-covered pixels, we used the MODIS cloud mask available at 1 km. The number of cloudy pixels were derived from state_1km band bits 0-1, which contains 4 classes: clear, mixed, not defined and cloudy. Only the cloudy class was selected to mark clouds on a glacier. As the cloud mask has a low resolution, the observations containing more than 3 cloudy pixels per glaciers are filtered out. Finally, the mean surface reflectance value in the NIR band over the glacier was calculated from cloud-free pixels.

#### SCAF S2

We derived the percentage of cloudy pixels over the glacier from the cloud mask, and land images with less than 10% of clouds were selected. To account for mountainous terrain, a topographic correction was performed. It is based on the Modified



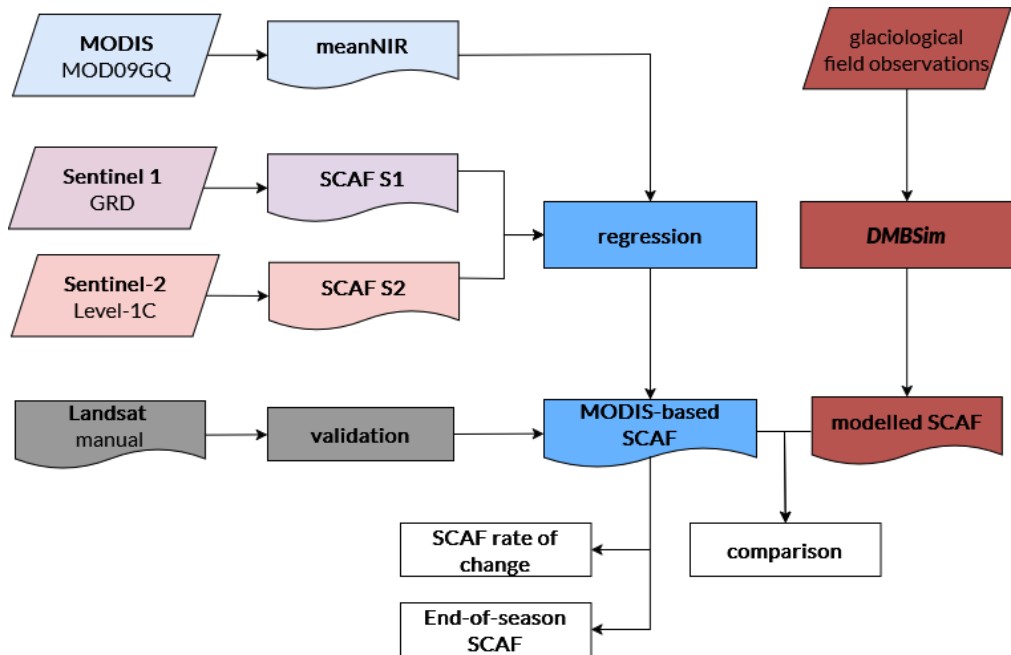

**Figure 2.** The workflow describing (i) MODIS-based SCAF time series generation by applying statistical relation to MODIS meanNIR and SCAFs derived from Sentinel-1 and Sentinel-2, (ii) extracting end-of-season SCAF and annual rate of change from MODIS-based SCAF time series and (iii) comparing the daily modelled SCAF from model to MODIS-based SCAFs

Sun-Canopy-Sensor topographic correction function developed by Soenen et al. (2005) and implemented in GEE by Macander et al. (2020).

### 3.1.1 Glacier surface classification with Otsu-threshold

The main purpose of the classification algorithm is to differentiate between snow-covered and snow-free surfaces. We used the NIR channel reflectance values and applied the Otsu-threshold algorithm (Otsu, 1979) to distinguish snow-covered areas
similar as in (Rastner et al., 2019). The Otsu algorithm is designed to find the optimal threshold (th) by maximizing the interclass variance between two classes on the image, assuming bimodal distribution on the frequency histogram. We defined the $th_{min}$ and $th_{max}$, as the range of validity for using threshold determined by Otsu algorithm. However, not all images have both snow and snow-free areas, where the frequency distribution is not clearly bi-modal. Thus, no optimal threshold can be identified. If the initially detected threshold lies outside of this range, a fixed threshold ($th_{fixed}$) is applied to classify the
image. The range $th_{min} < th < th_{max}$ as well as the fixed thresholds were manually identified after comparing classified images to the RGB composites of each glacier to improve classification (3). The found set of thresholds is applied to the entire time-series of pre-processed Sentinel-2 imagery.



Although ultimately $th_{fixed}$ must be provided, which makes the approach less automatic in that regard, the same set of iden-
tified threshold $th_{fixed}$, $th_{min}$,$th_{max}$ could be potentially applied to glaciers of the same basin for future regional application.

Finally, after the threshold is defined, all pixels with reflectance values above the threshold are classified as snow. SCAF S2
is calculated as a proportion of snow-covered pixels on the glacier to the total number of glacier pixels after the SLA correction
(see Sect. 3.1.2).

**Table 3.** Defined set of Otsu thresholds for all studied glaciers.

| Glacier | th_fixed | th_min | th_max |
| --- | --- | --- | --- |
| Abramov | 0.5 | 0.4 | 0.55 |
| Golubin | 0.54 | 0.4 | 0.58 |
| Glacier No. 354 | 0.45 | 0.35 | 0.5 |
| Zulmart | 0.45 | 0.4 | 0.55 |

*SCAF S1* During the melt season the liquid water content in the snow on the glacier increases, especially along the transient
snowline, steadily rising during the ablation season. The lower boundary of wet snow on a glacier detected on SAR images
can thus be used as a proxy for the transient snowline (Winsvold et al., 2016). In presence of water, high dielectric losses leads
to high absorption coefficient. The penetration depth in SAR C-band decreases from around 20 m in dry snow condition to
a few centimeters for wet snow, resulting in a low backscatter intensity (Nagler and Rott, 2000; Marin et al., 2020; Buchelt
et al., 2022). Such contrast in the backscatter intensity of wet snow and snow-free/dry snow-covered surfaces is the basis of
the well-established approach inNagler et al. (2016) to detect wet snow. The average backscatter ratio (R) image is obtained as
follows:

$$R_{\text{VV}} = \sigma_{\text{VV}} - \sigma_{\text{VVref}} \tag{1}$$

$$R_{\text{VH}} = \sigma_{\text{VH}} - \sigma_{\text{VHref}} \tag{2}$$

$$R = (R_{\text{VV}} - R_{\text{VH}})/2 \tag{3}$$

, where $\sigma_{\text{VV}}$ (dB) is the image backscatter intensity in VV polarization, $\sigma_{\text{VV\_ref}}$ is the median of reference images during the
winter season. This is similar for VH polarization. The final ratio image R is used to discriminate wet snow on the glacier.

To decide whether the pixel contains wet snow or not, a threshold is applied to the ratio image (R). We found a threshold of
-6 dB to be suitable. The binary images are resampled to 50 m to reduce the noise, and proportion of snow-covered area relative
to the total glacier area (SCAF S1) is calculated after the SLA correction (Sect. 3.1.2).

### 3.1.2    Snowline altitude correction

We performed SLA correction for the classified images of Sentinel-2 and Sentinel-1 in Sect. 3.1 based on the assumption that
pixels clearly above SLA are more likely to be snow covered than snow-free and vice versa. SLA detection adopts the altitude
bin approach implementation after (Zeller, 2020), which was initially developed by Rastner et al. (2019). The elevation range





of the glacier is divided into altitude bins, and for each bin the proportion of snow pixels is calculated. Due to the difference in glacier sizes, a bin size of 30 m is used for Abramov and a bin size of 25 m is used for the other three glaciers. The algorithm

searches for three consecutive altitude bins with at least 50% of snow cover, and takes the altitude of the lowest bin out of the three as SLA. Often the snowline is rather a transition zone where the glacier surface grades from snow, to snow patches, to bare ice, influenced also by superimposed ice and no clear-cutting line. Typically, slush, soaked snow or refrozen ice mark the transition (Cogley et al., 2011; Barandun et al., 2015). To respect this transition from ice surface to snow cover, we have defined a critical SLA range just above and below the measured SLA, for which no correction is applied. This range is established by

buffering the detected SLA both upward and downward by 20% of the difference between the glacier's maximum and minimum elevations, as snow(ice) is situated above(below) the detected SLA. Finally, pixels outside this transition zone are reclassified accordingly, and SCAFs are calculated.

### 3.1.3 $glacier SCAF_{MODIS}$ regression

We combined the time series of SCAF S1 and SCAF S2, where the priority was given to Sentinel-2 images if the dates were

overlapping. As the Sentinel-1 is unable to detect the end-of-season SCAF due to the limitation to detect dry snow (see Section **??**), we removed SCAF S1 values after mid of August for each year. To build a regression model, we selected all the dates with observations available for both MODIS meanNIR and SCAF S1 & S2 combined time-series. Initially we tried to fit the linear function; however, we found that the relationship between MODIS meanNIR and SCAF from high resolution satellites could be approximated better with the decay exponential function constrained by maximum possible SCAF of 100%:

$$y = (1 - a * e^{-b*(x)}) * 100 \tag{4}$$

We used an exponential regression between meanNIR as an independent variable (x) and combined SCAF S1 & S2 time series as a dependent variable (y). The coefficients a, b and c determine the shape of the curve. We derived the exponential regression for all four studied glaciers (Fig.3). The regression function was then used to derive SCAF from MODIS time series for the period before high-resolution data became available. The MODIS-based SCAF was validated with manually detected

Landsat-based SCAFs (see Sect. 4.1.2).

### 3.2 End-of-season SCAF and rate of change calculations

From SCAF time-series from the $glacier SCAF_{MODIS}$ algorithm, we excluded values that deviated by more than 1.5 standard deviations below the annual mean SCAF, in order to reduce the influence of anomalously low observations. For Glacier No. 354, no filtering was applied due to the limited number of observations and the frequent occurrence of summer snowfall, which

together increase the uncertainty in reliably identifying anomalous values. For each year, we fetched the minimum SCAF observation during the entire season for each corresponding glacier as end-of-season SCAF.

To study the interannual variability of the snowline, we calculated SCAF rate of change for summer months, that shows how fast the snow cover changed during the season after Pelto (2011) . The rate of change was calculated separately for each year.




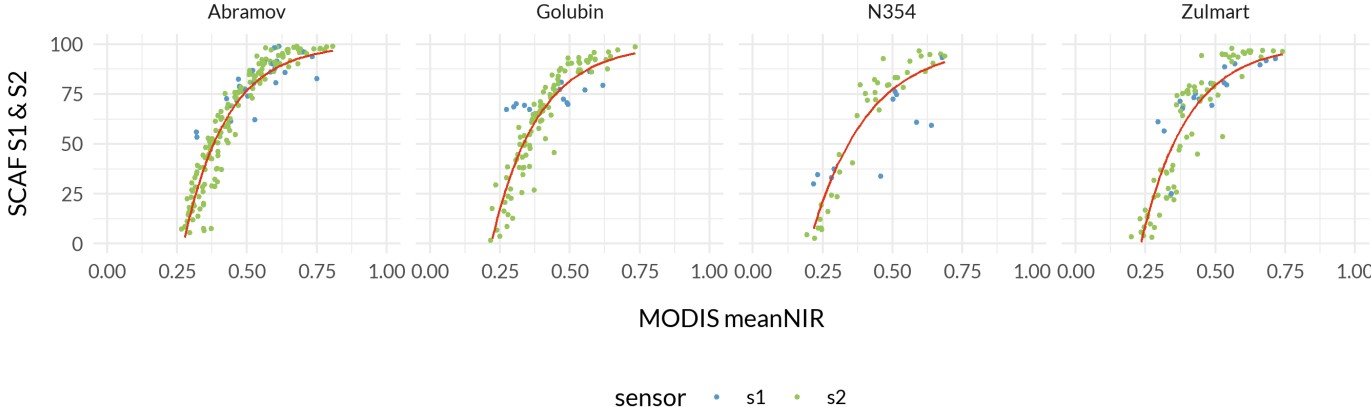

**Figure 3.** Exponential regression function to obtain SCAF for four study glaciers

To reduce the impact of the mid-season snowfalls we filtered SCAF time series for each year with filtering algorithm described
in Machguth et al. (2023), modifying it to detect the SCAFs higher than than the previous value. We applied the filter for all
glaciers and applied a linear regression to calculate the slope. We calculated the trends in annual snowline rate of change for
2000–2023 and 2009–2023 using Mann-Kendall test (Mann, 1945; Kendall, 1948).

## 4 Results

### 4.1 $glacierSCAF_{MODIS}$ sensor comparison and validation

#### 4.1.1 SCAF retrieval sensor comparison

We obtained SCAF values at four study glaciers during the melt season over the period 2000 to 2023 (Fig. 6). The proposed
methodology has demonstrated a notable capacity to represent the evolution of the SCAF on a glacier during the entire melt
season at close to daily resolution, also for periods before high-resolution sensors were available (e.g. 2008 in Fig.4). Snow
depletion over the glacier surface occurs typically between June and September. The snowline rises until it reaches the end-of-
summer SCAF, followed by fresh snowfall that remains during the winter season. $glacierSCAF_{MODIS}$ showed comparable
performance to the SCAF obtained from high-resolution satellite imagery throughout the season for all study glaciers from
2015 to 2023 (e.g. year 2018 in Fig.4). Moreover, with the high density of recorded SCAFs (Fig.6), it is possible to detect
mid-season snowfall (e.g. year 2008 for Abramov DOY 250; year 2018 for Golubin DOY 243 in 4), when SCAF increased
rapidly for a short period and declined again.
While MODIS-derived SCAFs were often lower than Landsat-based manual delineations (Fig. 4.1.2), the higher temporal
resolution of MODIS allowed a more accurate capture of the end-of-season values. For example, recorded end-of-season
SCAFs from manually mapped Landsat imagery were 59% in 2008 for Golubin Glacier (DOY 211), while the snow cover





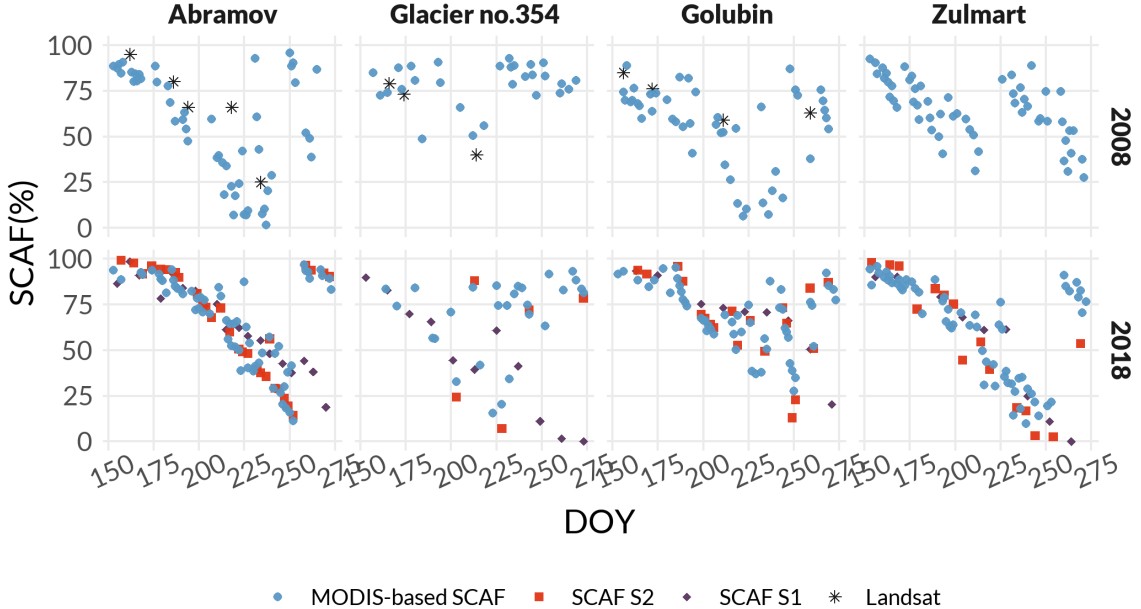

**Figure 4.** Glacier SCAF evolution during the melt season derived from MODIS, Sentinel-1, Sentinel-2 and manually from Landsat imagery.

clearly continued to deplete and the highest position of the snowline occurred later in the season corresponding to a SCAF of 18.7% (DOY 236, Fig.4).

Sentinel-1 SCAFs retrieved before the mid of August are on average higher than MODIS-based SCAFs by 2.46% and by 7.13% in absolute difference for Abramov, Golubin and Zulmart glaciers. For Glacier No. 354 the MODIS-based SCAFs were higher than derived from Sentinel-1 by 2.44% and 12.10% in absolute values. If the entire period is taken into account, the MODIS-based SCAFs are higher than Sentinel-1 SCAFs by 25% (abs:30.45%), followed by Zulmart at 12.78% (abs:20.82%), and Abramov Glacier 0.43% (abs" 13.14%), while for Golubin Sentinel-1 showed higher SCAFs on average by 0.82% (abs:19.27%). The clear example of Sentinel-1 and MODIS-based SCAF disparity towards the end of the season could be observed on Fig.4 for Glacier No. 354 in 2018. The SCAF difference between Sentinel-2 and MODIS-based SCAFs is ±1% on average and 7.60% in absolute values for all glaciers with largest absolute value for Zulmart glacier (9.8%).

### 4.1.2 SCAF validation with Landsat-derived snowlines

We compared the SCAFs derived from $glacierSCAF_{MODIS}$ with the manually derived SCAFs from Landsat (Fig. 5). Abramov Glacier, Glacier No. 354 and Golubin Glacier exhibited an RMSE below 20%. For these three glaciers, the $glacierSCAF_{MODIS}$ results are systematically underestimated by an average of 16.3%. Visual inspection of corresponding Landsat scenes suggests that the MODIS-based SCAFs tend to underestimate snow cover, particularly within the 50–75% SCAF range. Zulmart glacier



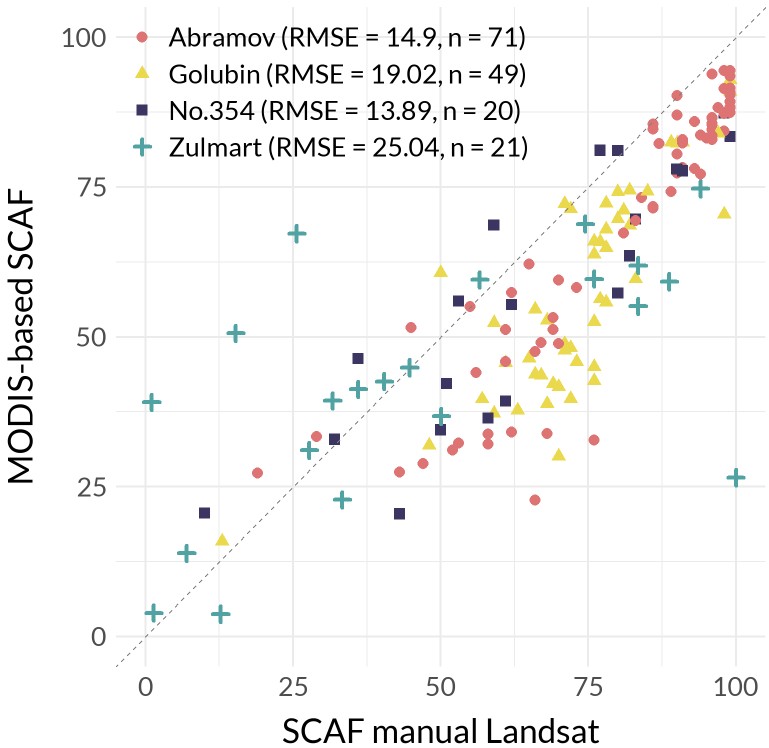

**Figure 5.** Independently delineated snowlines from Landsat imagery compared to the MODIS-based output

showed less good agreement, with a slightly higher RMSE of 25%. SCAFs in the lower ranges (between 0-50%) are sporadically overestimated on MODIS images on average by 19.7%.

## 4.2 $glacier SCAF_{MODIS}$ time series

### 4.2.1 Long-term snowline temporal dynamics at sub-seasonal scale

More than 20 years of the snowline observations on an almost daily scale are displayed in Fig.6. In June, the SCAF of Abramov and Zulmart glaciers ranges between 75-100% throughout the study period. This is 10-15% higher than at the glaciers Golubin and Glacier No. 354, where the ablation season starts earlier.

In 2001, at least 75% of the Abramov surface remained snow-covered until early August; however, in 2023, this same proportion of snow cover was already observed in mid-July. Over the past 20 years, the snow cover on Golubin and Glacier No. 354 has been depleting more quickly, with less than half of the glacier remaining snow-covered by mid-July in 2023, whereas previously such levels of snow cover was typically reached in late July or early August. For Zulmart, no consistent changes were found in the beginning and middle of the melt season. We observed intensive melt (SCAF < 15%) continuously





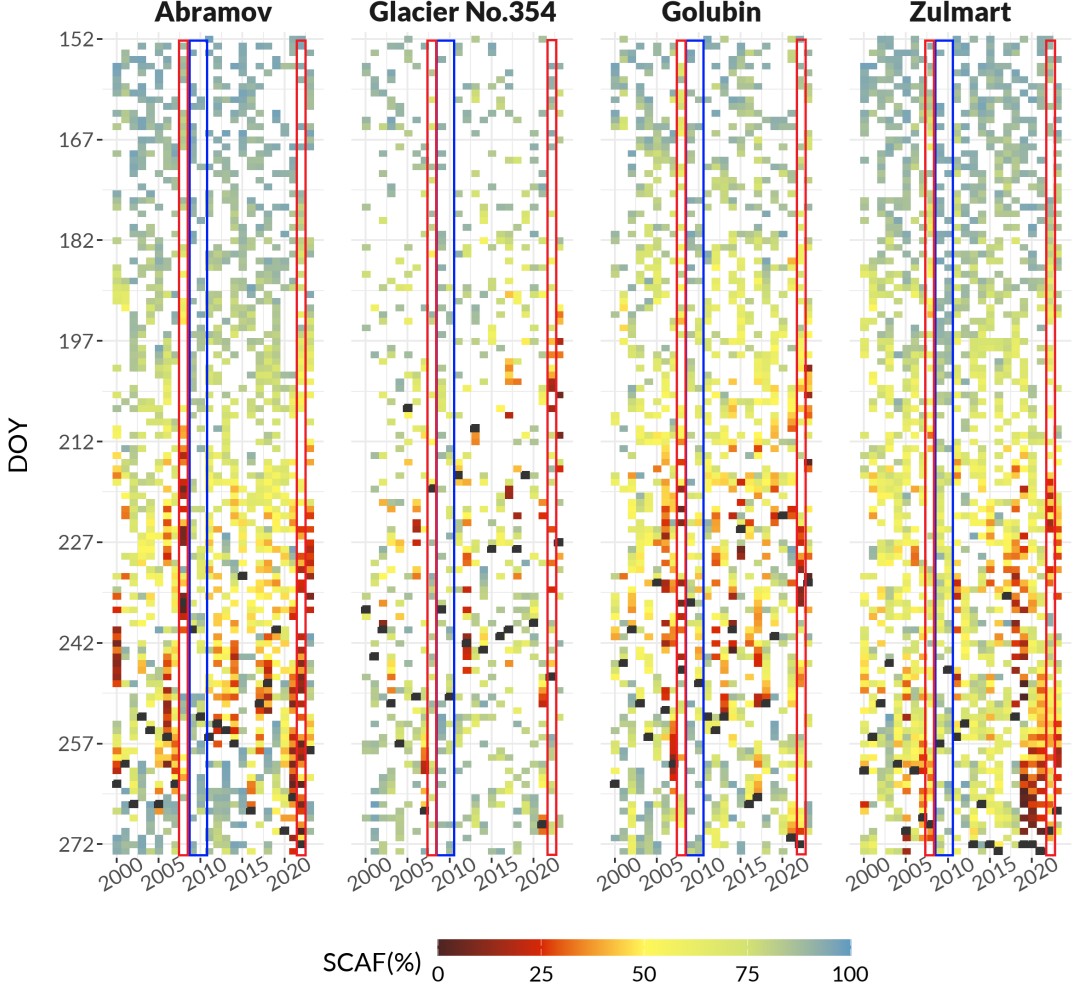

**Figure 6.** Heatmaps of the MODIS-based SCAF values for four glaciers, 2000-2023. The end-of-season SCAF is marked in the black boxes. The blue and red rectangles highlight the years 2009 and 2010, and 2008 and 2022 respectively (Sect.4.2.1).

until the end of September, while in 2012-2016 the season ended earlier at the end of August with SCAF of at least 30%. The end-of-season SCAFs are described in detail in Sect.4.2.2

In 2009 and 2010 all the glaciers maintained half of their snow cover throughout the entire season (Fig. 6, the blue box). Conversely, in 2018 and 2023 the snowline on all study glaciers rose rapidly to higher altitudes almost completely exposing ice and firn surfaces on the entire glacier. Similarly, in 2008 and 2022, SCAF fell below 10% for Abramov, Golubin and Zulmart

glaciers (Fig. 6, the red box). Although both years exhibited extreme snow depletion, in 2008, the melt season ended at the beginning of August, whereas in 2022, it persisted longer, until end of August and the beginning of September. The length of the season was notably extended in 2021 for Abramov, Glacier No. 354, and Zulmart. At the same time at Golubin glacier,




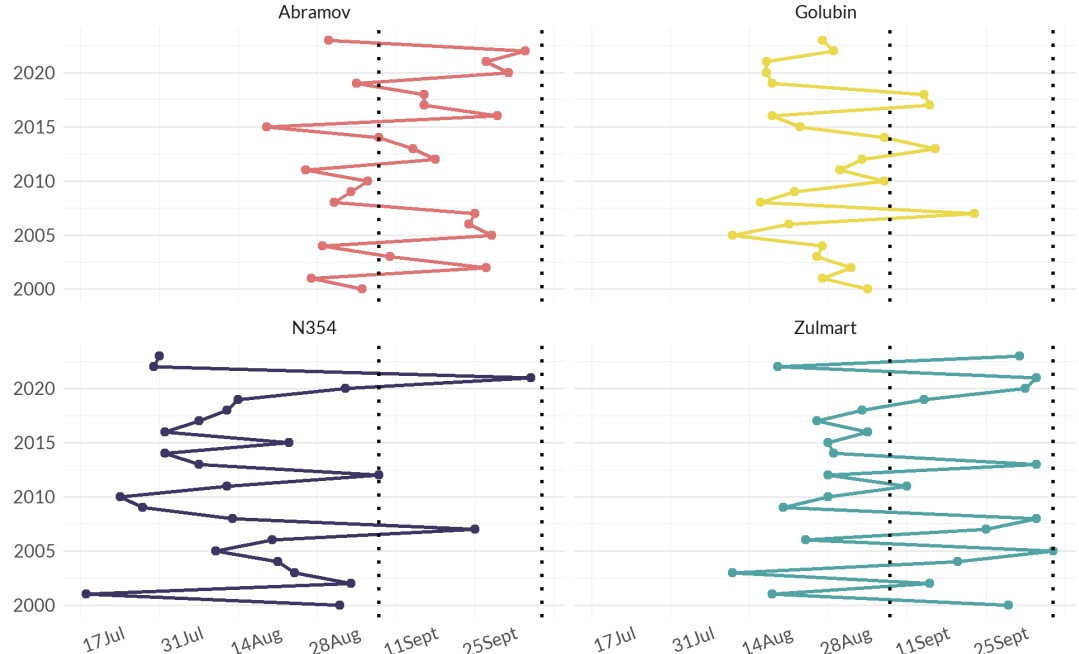

**Figure 7.** End-of-season snowline for four glaciers. The dashed line indicate beginning and end of September.

there were multiple mid-season snowfalls in July and August when the snowline dropped briefly and then rose again as ablation continued.

### 4.2.2 End-of-season SCAF dynamics and trends in annual rate of change

? suggested that the timing of the end of season is more or less restricted to last 10 days of September, in contrast, at least in case of the Abramov Glacier, we observed the end of ablation season date fluctuating between end of August and September (Fig.7). Despite relatively high inter-annual variability, a trend to a later end of the melt season has become visible since 2009. Surprisingly, the ablation season for Golubin, the lowest-lying of the four glaciers (Table 1), ends in the first half of August, with the lowest year-to-year variability and tends rather to an earlier onset of the winter season since 2009. Glacier No. 354 and Zulmart, both located at higher elevations and influenced by continental climatic conditions, show now clear trends and high interannual variation in the timing of the end of the ablation period. While for Glacier No. 354 the ablation season often ends early in August, it tends to systematically extend into September for Zulmart.

We found the annual snowline rate of change for 24 years fluctuates between -0.2 and -1.25%/day for all the study glaciers, except for Glacier No.354 as the SCAF depleted faster reaching -2%/day in 2014 (Fig. 8). A negative rate of change signifies a decrease in snow cover with larger negative values indicating a more rapid depletion of snow on the glacier over short period of time. Glacier No.354 exhibited higher variability especially in the second decade in comparison to the other three glaciers. The





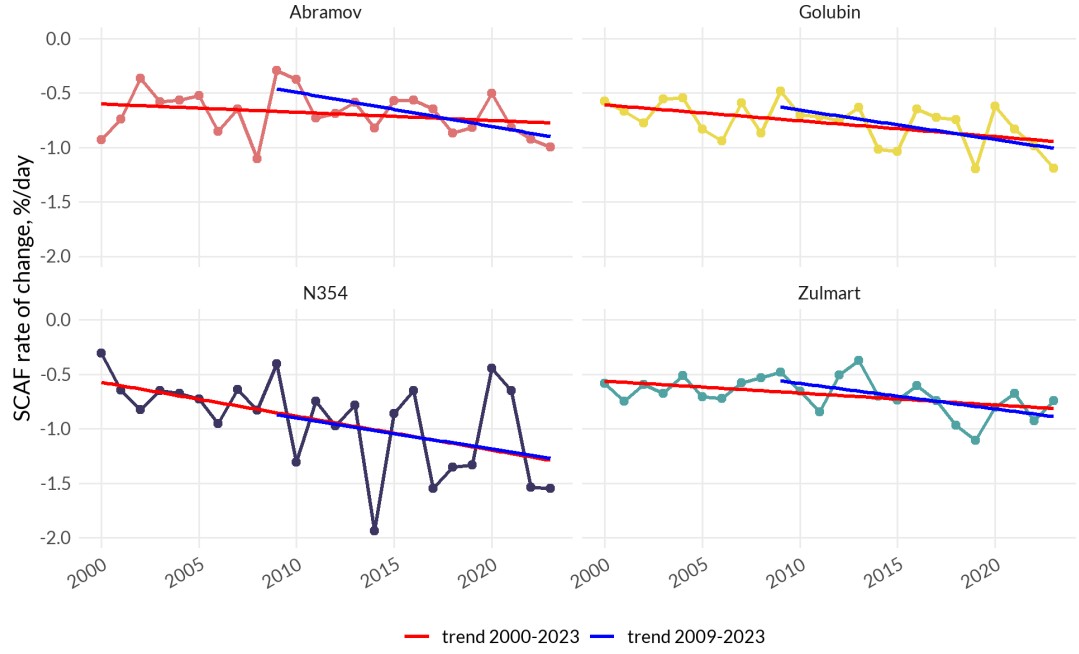

**Figure 8.** Annual snowline rate of change derived from MODIS-based SCAF.

Mann Kendall test revealed the decreasing trend at 0.05 significance level for Golubin, Glacier No. 354 and Zulmart between 2000 and 2023 (Table 4). From 2009, an significant accelerating trend of snowline retreat is also found for the Abramov glacier.

**Table 4.** Mann-Kendall trend results for SCAF rates of change over two periods (2000–2023 and 2009–2023) for four glaciers. An asterisk (∗) indicates statistical significance at the 95% confidence level (p < 0.05)

| Glacier | Period | p-value | Kendall's tau ($\tau$) |
|---|---|---|---|
| Abramov | 2000-2023 | 0.22421 | -0.18 |
| | 2009-2023 | 0.02282 | -0.49* |
| Golubin | 2000-2023 | 0.01845 | -0.35* |
| | 2009-2023 | 0.03767 | -0.41* |
| Glacier No. 354 | 2000-2023 | 0.01062 | -0.38* |
| | 2009-2023 | 0.37305 | -0.18 |
| Zulmart | 2000-2023 | 0.04452 | -0.30* |
| | 2009-2023 | 0.03767 | -0.41* |





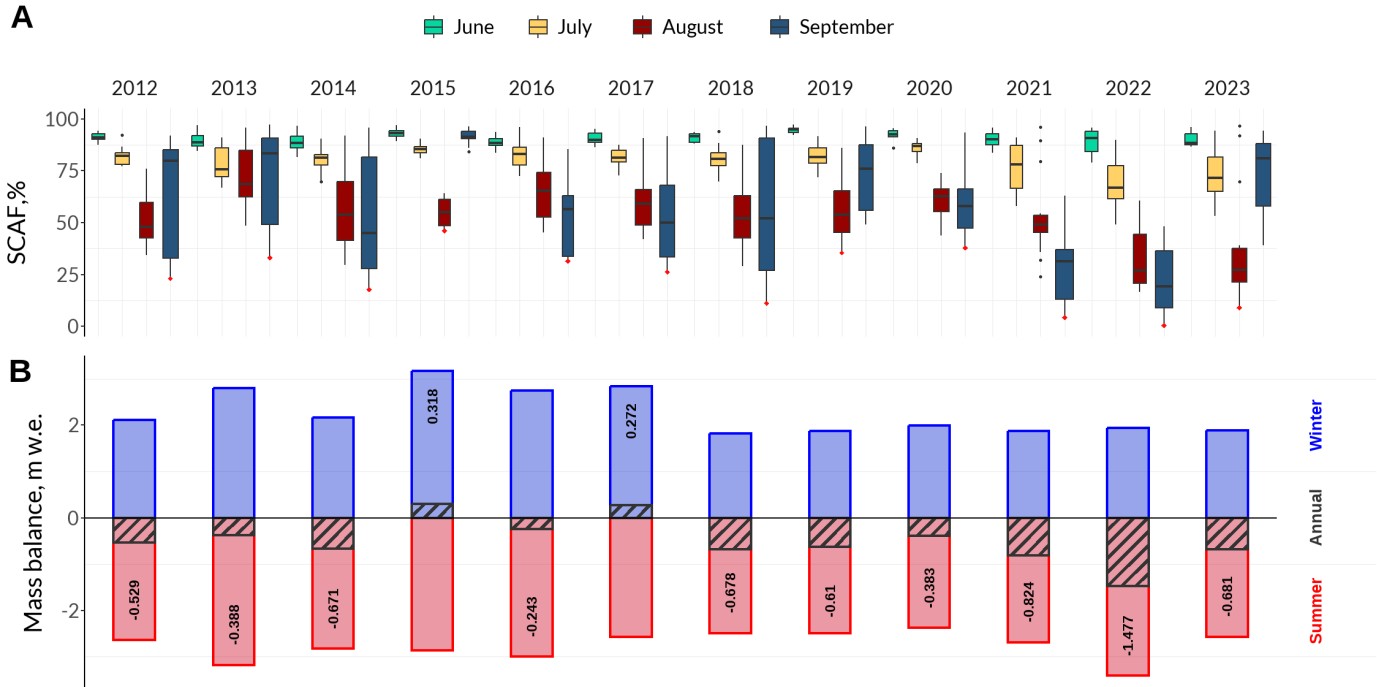

**Figure 9.** Panel A shows monthly SCAF as a boxplot for Abramov glacier. The red point is the end-of-season SCAF. Panel B displays the corresponding measured annual mass balance (dashed line), accumulation (blue), and ablation (red).

## 4.3 SCAF in relation to the mass balance

### 4.3.1 SLA / SCAF and measured annual mass balance

We compared mean monthly SCAF and monthly variability against measured annual mass balance, rather than relating end-of-season SLA/SCAF to annual surface mass balance. While we did not find a clear relationship with length of ablation period or monthly SCAF, we identified a complex relationship between sub-seasonal SCAF variability and annual surface mass balance (Fig.9). Here we analyzed the results for the Abramov glacier with the longest measurement time series.

In recent years, the surface mass balance for the Abramov glacier has become more negative, as reflected also in the SCAF (Fig. 9, B). Since 2021, the SCAF has dropped in comparison to previous years during all summer months. However, the observed monthly SCAF changes are not linear. For instance, the end-of-season SCAF (mean September SCAF and its variability) in 2021 and 2022 were very similar; however, the difference in surface mass balance was significant ($\Delta$ 0.65 m w.e.). This difference between the two years might be explained by a different glacier response in July and August of the two years, reflected in the SCAF dynamics. In 2021, snow cover persisted at lower elevations for a longer period during the ablation season (higher SCAF for July and August). At the end of the season, rapid snowmelt exposed darker ice, which was accompanied by a sharp





rise in the snowline and a drop in SCAF in September to levels similar to those in 2022. Thus, the highly reflective snow cover, which persisted for a longer period during the season, protected the glacier ice from melting and reduced the annual mass loss

of the glacier. In comparison, for 2022, the SCAF was very low already in July and August leaving the spectrally darker ice at the surface earlier in the season than usual. The snow cover depleted at a similar rate from June to August in 2023 as in the extreme negative mass balance year of 2022. However, the melt season ending earlier in September resulted in more favourable conditions than in 2022. Several fresh snow fall events in late August/early September protected the ice from further melting (Fig. 6).

**4.3.2 SLA / SCAF for model validation**

We used the $glacierSCAF_{MODIS}$ time series to validate the modelled SCAF from *DMBSim* (Sect. 2.3.5) for all four glaciers (Fig. 10). It provides a fast and less resource-intensive and unique way to evaluate the performance at sub-seasonal scale of the surface mass balance model calibrated with annual glaciological observations. In the following we describe in detail the application of $glacierSCAF_{MODIS}$ in combination with mass balance modelling for the Abramov glacier.

The modelled SCAF during summer months shows snow depletion pattern similar to the retrieved time series from MODIS (Fig. 10). For the Abramov glacier, summer snowfalls as well as the end of season is well approximated by the model. Drops in air temperature, often show a slow down of SCAF changes and often relate to summer snowfall events (Fig. 10 e.g. 2014). Despite the limited accuracy of the precipitation dataset (ERA5) (Zandler et al., 2019; Dollan et al., 2024), the model captured well the seasonal snowline evolution during the summer. However, we noticed two important tendencies in the predicted

modelled daily SCAF: 1) the modelled SCAF is overestimated by the model, especially in earlier years and 2) snow depletion tends to be too fast and abrupt during July to August (critical period for meltwater release).

While in the model, melt rates are low at the beginning of the season, they accelerate in mid July and peak at the end of July to mid-August, then drop again in September. However the evolution of the SCAF depletion observed on the remote sensing input is more gradual, especially for July and August, suggesting a more moderate meltwater production during the summer

420 months than simulated by the model.

To compare the depletion pattern between the modelled and observed SCAF, we manually defined the critical period when the snow depletes most rapidly for each year and filtered out snowfalls from both time series (Fig. 10, solid red and blue lines). This period is also crucial for water availability. Compared with observations, the model simulated a later start of snow depletion at the beginning of the season. This was followed by a faster melt during the critical period, which compensated

for the excess snow. Consequently, the model generally represented the dynamics of the snowline at the end of the season well. The discrepancy during this critical period is most apparent in 2020 when model melted out the snow almost two times faster (-2.02%/day vs. -1.1%/day). Furthermore, modeled fresh snowfall seems to deplete too fast and too strong after a fresh snowfall event (Figure **??**daily) e.g. end of season 2019). This shows that despite a good annual fit with direct mass balance point observations the seasonal components can be over / underestimated: thus too much winter snow is compensated with too

high melt rates and vice-versa.





**Figure 10.** The top panel shows MODIS-based SCAF (red) and modelled SCAF (blue) for the Abramov Glacier from 2012 to 2023. The modelled SCAF is calculated from the mass balance model closely calibrated to annual mass balance point observations (Sect.2.3.5). Model performance with glaciological point observations is provided by the RMSE for each year. Annually calibrated cumulative melt is shown as a black solid line. The bottom panel displays the average daily air temperature over the same period.



While it might not influence the annual estimate of surface mass balance change and melt water release, it becomes crucial for understanding meltwater input during the dry summer months and potential future runoff changes for Central Asia (Barandun et al., 2020).

## 5 Discussion

### 5.1 SCAF dynamics and sub-seasonal meltwater changes

Our findings suggest that similar annual surface mass balances can have very different seasonal distributions of SCAF. This needs to be considered for accurate meltwater predictions, particularly given the anticipated future changes in seasonal meltwater contributions during the dry summer months (Huss and Hock, 2018; Armstrong et al., 2019). This study provides more than 20 years of almost daily snowline observations on four remote glaciers. In combination with modelled sub-seasonal mass balance time series, it allowed us to link the key changes in close-to-daily glacier snowline dynamics and provide information on sub-seasonal monthly meltwater contribution changes for two glaciers located in the Syr Darya basin (Golubin, Glacier No. 354) and Amu Darya basin (Abramov), the two major rivers basins of Central Asia. In the following we highlight the key findings for each glacier and discuss the implication on meltwater contribution to the total river runoff.

The key changes in SCAF dynamics for the Golubin glacier suggests an earlier onset of the winter season since 2018 (Fig.7). This is surprising considering a more and more negative mass balance in recent years (Barandun et al., 2025a). We also observed a consistently increasing negative rate of SCAF migration over the study period (Fig. 8), meaning that the snow cover depleted more quickly, uncovering the underlying ice earlier in the season and making the glacier more susceptible to accelerated melting. More intense precipitation events are expected towards the end of August, when night-time temperatures already start to drop below zero, leading to an early, persistent snow cover that will reduce glacier melt again. Since the early 20th century, Golubin has experienced simultaneous increase in accumulation and ablation (an increased mass turnover and a steeper mass balance gradient), as suggested by Azisov et al. (2022). The same author highlight that such changes can be linked to a potential shift towards a more humid and warm settings that implies changes in the glacier regime and sub-seasonal meltwater dynamics Dyurgerov and Meier (2000); Meier et al. (2003). Our results suggest an earlier, more intense melt season with a decrease in meltwater release closer towards autumn. Similar observations have been made on Barkrak glacier in a similar setting as Golubin glacier at the Western margin of the Tien Shan (Appendix B1).

Similarly, snow melt intensified for Glacier No. 354, especially in July. There was no clear trend evident over time for the highly variable end-of-season timing (Fig.7). The glacier had the most variable and negative rate of SCAF change of the four investigated glacier (Fig. 8). A higher variability of the SCAF and the end of season timing can be explained by frequent summer snowfalls that have in the past significantly reduced melt rates (Kronenberg et al., 2016). The strong negative trend of the rate of change point towards significant changes in glacier response. The mass balance of Glacier No. 354 has been reported as increasingly negative in most recent years WGMS (2024); Barandun et al. (2025a) with values higher than the average mass loss of the glaciers in Central Tien Shan Kenzhebaev et al. (2017); ? and the regional average Hugonnet et al. (2021); Rounce et al. (2021). Glacier response has, however, been reported to be very heterogeneous in the region Barandun



et al. (2021). This could so far not be explained by climatological or topo-morphological drivers due to the lack of accurate data
(Barandun and Pohl, 2023). Currently observed increased summer temperature lead to more frequent rainfall events in early spring and summer (M. Barandun, personal communication, 2025). In addition, mining activities in the immediate vicinity have accelerated and have affected nearby glaciers in the past Evans et al. (2016). Collectively, these interacting processes and feedback mechanisms complicate the accurate assessment of current state and the projection of future changes in glacier mass balance and runoff at seasonal to sub-seasonal scales in the dry, cold climate of the Central Tien Shan.

For the Abramov glacier, the melt season tends to end later in the year in most recent years. Over the past three years, the mean SCAF position has dropped for all summer months and its interannual variability increased substantially in July and even slightly in June. This also indicates an earlier onset of the melt seasons in addition to the higher meltwater release at the end of summer. Due to this prolonged melt season (Fig. 7) and the generally higher positions of the snowline during all summer months, the rivers swell earlier and for a longer period of time (Fig. 9. This is consistent with more and more negative
annual mass balances observed for the Abramov glacier Mattea et al. (2025); Saks et al. (2024); Kronenberg et al. (2022); Barandun et al. (2025a). According to the observed monthly snowline dynamics from 2000 to 2023 (Fig. 9), biggest year-to-year fluctuation of meltwater release occurred in August and September. While the high variability in September is largely due to an earlier or later onset of persistent snowfall, the variability in August is more likely linked to changes in air temperature, directly and indirectly influencing the glacier melt (i.e. inverse relation to the bare-ice albedo (Volery et al., 2025)).

The key finding for Zulmart is the abrupt change that has occurred since 2018. Since then, the depletion of snow cover on Zulmart has notably accelerated, particularly in September, with weaker changes observed also in August. Changes towards the end of the summer season in recent years have also been reflected by increased year-to-year variability in the SCAF in August and September, in contrast to decreased variability in June and July. Although meltwater release appears to be more consistent from year to year in early summer, predicting the end of the melt season and its magnitude has become more challenging.

Glacial melt is a major contributor to the Syr and Amu Darya rivers in Central Asia. During the dry season from July to September, it provides up to 70–90% of their total runoff (Armstrong et al., 2019). Future model simulations (Bosson et al., 2023; Zekollari et al., 2024) using a similar mass balance model setup to that used here predict that the contribution of glacier meltwater to the main rivers will decrease within the next few decades for the Syr Darya and only beyond the end of the century for the Amu Darya (Barandun et al., 2025b). The models also predict significant seasonal changes, such as an earlier onset
of glacier melt in spring across the entire region, and a general decrease in runoff in late summer for the Syr Darya. These predictions are partly consistent with our findings. However, we also found that such simple models don't focus on accurately reflecting sub-seasonal mass balance and melt distribution, especially when not calibrated on a sub-seasonal to daily scale. Seasonal to sub-seasonal results must be interpreted with great care, even when model calibration and forcing are based on in situ observations. For example, we found that modelled melt rates tend to be low at the beginning of the season, accelerate
in mid-July, peak at the end of July to mid-August and then drop strongly again in September. However, the observed SCAF evolution suggests more gradual meltwater production during the summer months. Failing to capture the correct sub-seasonal melt distribution, even when measurements are available, implies large uncertainties in the prediction of future runoff changes. Seasonal shifts have, however, extreme consequences for the irrigation sector and must be accurately captured for improved





water resource management (**?**). Our results thus highlight the importance of carefully assessing the sub-seasonal glacier
response, as well as the need for accurate observational data at seasonal to close-to-daily timescales to improve water resource
management.

### 5.2  $glacierSCAF_{MODIS}$ performance

#### 5.2.1  Snowline mapping for different glacier surface characteristics

The exponential relation between MODIS meanNIR over the glacier and SCAF S1 & SCAF S2 persists for all the studied
glaciers (Fig. 3). Further testing on other glaciers in the region revealed the same exponential relationship (Appendix B1) and
underlines the robustness of the here proposed methods. The seasonal snowline time series thus represent the snow depletion
patterns on the glacier during the melt season (Fig. 4).

   However, limitations related to the different glacier surface characteristics must be taken into the consideration. For example,
below the snowline, refrozen meltwater of seasonal snow contributes to the accumulation and moves the ELA lower than the
transient snowline. This zone of superimposed ice (e.g Abramov glacier, Glacier No. 354) is, however, difficult to detect via
optical spaceborne imagery Kundu and Chakraborty (2015). For Zulmart glacier the overnight refrozen ice is much brighter
spectrally, which makes it difficult for manual evaluation. The optical reflectance for the bright ice in NIR range is however
distinctive from the snow and, therefore, possible to separate Naegeli et al. (2017) with the selected Otsu threshold to be more
sensitive than for glaciers with mostly dark ice (Appendix **??**).

Over the past decade the $glacierSCAF_{MODIS}$ detected snowlines rising above the glacier. In this case, the older firn from
previous years is exposed and detected as a no-snow class. However, the mis-classification with younger firn is possible during
the rise of the snowline due to the higher reflectance. As Aberle et al. (2025) pointed out, using the separation of the snow and
firn class is better in the Sentinel-2 Surface reflectance product we used (Sect. 3.1) than in the Top of Atmosphere product.

#### 5.2.2  Uncertainties and limitations

The core of the proposed method is based on the regression between the mean NIR reflectance extracted from MODIS and
SCAF from Sentinel-2 and Sentinel-1 high-resolution satellites. We found the decay exponential function (increasing form)
to best approximate the relationship between two variables (Fig. 3) . Naturally, the lower MODIS-based SCAF values are
more sensitive to the small changes in the meanNIR reflectance. The end-of-season values below 20% are subject to higher
uncertainties than SCAF values during the season.

$glacierSCAF_{MODIS}$ performance depends primarily on quality of MODIS cloud mask and the accuracy of mapped snow-
lines from Sentinel-1 and Sentinel-2. For persistently cloud covered glaciers, such as Glacier No. 354, the less frequent obser-
vation in combination with low reliability of cloud mask result in blurred end-of-season snowline, and the snowline dynamics
of the glacier in such condition is more challenging to resolve.

   The NIR band of MODIS is available at 250 m resolution, but the cloud mask has a resolution of 1 km, which is coarse for
glacier scale applications and limits the benefit of implementing higher resolution MODIS imagery (Luo et al., 2008). Also,




poor performance of MODIS cloud mask is reported for snow-covered regions due to confusion between clouds and snow classes (Stillinger et al., 2019), where many snow pixels were mistaken for cloud cover, reducing a number of potentially usable imagery. The reverse case with high omission error could lead to the SCAF overestimation in the MODIS-based time-series and affects the short summer snowfall detection in case the thin snow-cover rapidly melts out. The heavier summer

snowfalls are less affected, as snow would melt during several days.

We mapped SCAFs automatically from both Sentinel-2 and Sentinel-1 sensors based on the established threshold classification techniques for optical (Otsu, 1979) and radar imagery (Nagler et al., 2016). Our results are affected by sensor specific limitations as well as the selected thresholds. In addition to the cloud cover, Sentinel-2 imagery is affected by cloud shadows on the glacier and the steep topography (e.g Golubin) that lower the SCAF S2 values, as the areas are classified as snow-free.

The similar issues are described by Rastner et al. (2019). To partially correct it, we applied SLA approach to reclassify snow pixels in the ablation area and no-snow pixel in the accumulation zone.

Although Sentinel-1 images are not affected by clouds, dry snow is transparent in the C band (Nagler et al., 2016), which excludes accurate capture of summer snowfalls and end-of-season snowline. Also, the fixed threshold of -6 dB for wet snow detection is applied, which is not necessarily optimal for all the study glaciers throughout the ablation season due to the

differences in climatic settings, sensor viewing angle and polarization (Winsvold et al., 2018; Buchelt et al., 2022). Sentinel-1 is able to provide valuable information on the snowline observations by itself, but as $glacierSCAF_{MODIS}$ is optical in nature, the potential of Sentinel-1 imagery has not been fully realized.

Our results were validated against manually delineated SCAFs from Landsat, which is subject to evaluators interpretations on snow and ice facies classification and underlies also considerable limitations, especially for different observers. Throughout

the entire study period we used fixed glacier outline from GLIMS and RGI v6, which underwent inevitable changes (Khromova et al., 2014). As glaciers retreat, the fraction of snow within fixed glacier area artificially reduces over time. Additionally, the geometry of the glacier and the orientation of its tongue changes over time.

## 5.3 SLA / SCAF as a proxy for the mass balance

Challenges remain to use the end-of-season snowline for the ELA/AAR concept despite the dense time series of SLA/SCAFs

(Dyurgerov, 2010).Letréguilly and Reynaud (1989) showed that mass balance variability correlates over distances of a few hundred kilometers and hold for glaciers that are located under fairly different climatic settings. However, alliance between the different parameters is glacier specific and hence an extrapolation of an established relationship between the ELA/AAR and the SLA/SCAF for individual, well observed glacier to other glaciers connects to large uncertainties. Our results show varying strengths of the relation between the annual mass balance and SLA/SCAF of the investigated glaciers (Appendix **??**). This

relationship depend on a range of different site-specific conditions.

The remotely sensed snowline detection can be interrupted by summer snow events. These summer snow events can have two effects on glacier mass balance. On the one hand, it can contribute to accumulation if it is significant enough (Fujita, 2008). On the other hand, it increases the surface albedo and thus reduces melt rates (Kronenberg et al., 2016). While after small events, the freshly fallen snow melts rather quickly and the previous (seasonal) snowline is restored, large events can last





several days and change the location of the transient snowline over a longer period or permanently for the remaining ablation period. In such cases, it is unclear to what extent the SLA/SCAF is an accurate reflection of the ELA/AAR. A simple end-of-season relationship between SLA/SCAF and mass balance may therefore not correctly reflect the annual glacier mass balance. Glacier No. 354, a glacier prone to prolonged summer snowfall, is a good example. Similarly, Dyurgerov (2010) found no clear relation between SLA and mass balance for Kara-Batkak glacier, also located in the Central Tien Shan. Another complication
blurring the SLA/SCAF and mass balance relation is superimposed ice.

An additional problem for the use of the ELA concept is the rise of snowline above the maximum altitude of the glacier. Studies such as Huss et al. (2013) and Hulth et al. (2013) have changed the use of SLA/SCAF time series from a direct proxy for mass balance to focus on the information stored in the transient snowline dynamics. In this way, a relationship not only with the position but also with the changes of the snowline during the ablation season and the surface mass balance can be
established, for example for model calibration (Barandun et al., 2018) or validation (Kenzhebaev et al., 2017; Kronenberg et al., 2016). This overcomes the shortcomings of inaccurate approximation of the end of the season or the snowline rise above the glaciers and renders the application less sensitive to a single observation (Barandun et al., 2018). In Barandun et al. (2021), Landsat and ASTER based snowlines were used for direct model calibration, showing a potential to improve region-wide mass balance modelling. However the authors also state that sub-seasonal mass balances are still very uncertain due to the few
transient snowline observations available during the summer months, especially at the onset of the 21st century.

The high-resolution time series of snowlines presented here offer unique insights at the sub-seasonal time scale, facilitating model calibration or validation (Sect. 4.3.2) (Barandun et al., 2018, 2021). The dataset can bridge the lack of glaciological measurements or complement annual observations with sub-seasonal information, which are scarce for the Tien Shan and Pamir. Providing close-to-daily SCAF/SLA observations therefore allows for a more accurate estimation of sub-seasonal/daily
mass balances. This helps us to better understand the sub-seasonal response of glaciers to climate change, and the subsequent release of meltwater into Central Asia's major rivers, which are a vital source of freshwater during the dry summer months (Barandun et al., 2020; Armstrong et al., 2019; ?).

## 6 Conclusion

The present work introduces the algorithm to infer the glacier snowlines at high temporal revisit based on multi-resolution
satellite imagery. Using the relation between the glacier surface reflectance and fractional snow cover derived from high-resolution imagery, we reconstructed snowline time-series on close-to-daily scale since early 2000s, including the period with no glaciological observations in Tien Shan and Pamir mountain ranges.

We found glaciers in the Tien Shan exhibit increasingly heterogeneous snowline dynamics in both timing and magnitude, whereas glaciers in the Pamir have responded more uniformly in recent years. These findings align with previous studies
focusing on glacier mass balance response to climate change and further emphasize the spatial complexity of glacier-climate interactions in Central Asia. Our observations are in agreement with snowline maps from high-resolution satellite and manual delineations from independent sources. Although uncertainties remain due to sensor limitations e.g. snow/firn discrimination,



or cloud masking - the high temporal density of observations reduces the impact of individual outliers. The computational efficiency and scalability suggest good potential for regional applications. Our results provide consistent and independent data
on glacier state and dynamics over time.

Rather than relying solely on end-of-season snowlines to estimate annual surface mass balance, we demonstrate that sub-seasonal snowline evolution carries valuable information about intra-annual snow melt dynamics, and can be used to independently validate glacier mass balance models. This is especially relevant in regions where in-situ measurements are sparse or absent. We show that similar annual mass balances can result from different seasonal melt patterns, which has implications for
predicting glacier runoff during the critical summer months. Thus, we highlight the importance of sub-seasonal glacier melt component distribution in future water availability scenarios under changing climate in the Central Asian mountain ranges.

## Appendix A: Study sites' satellite scenes

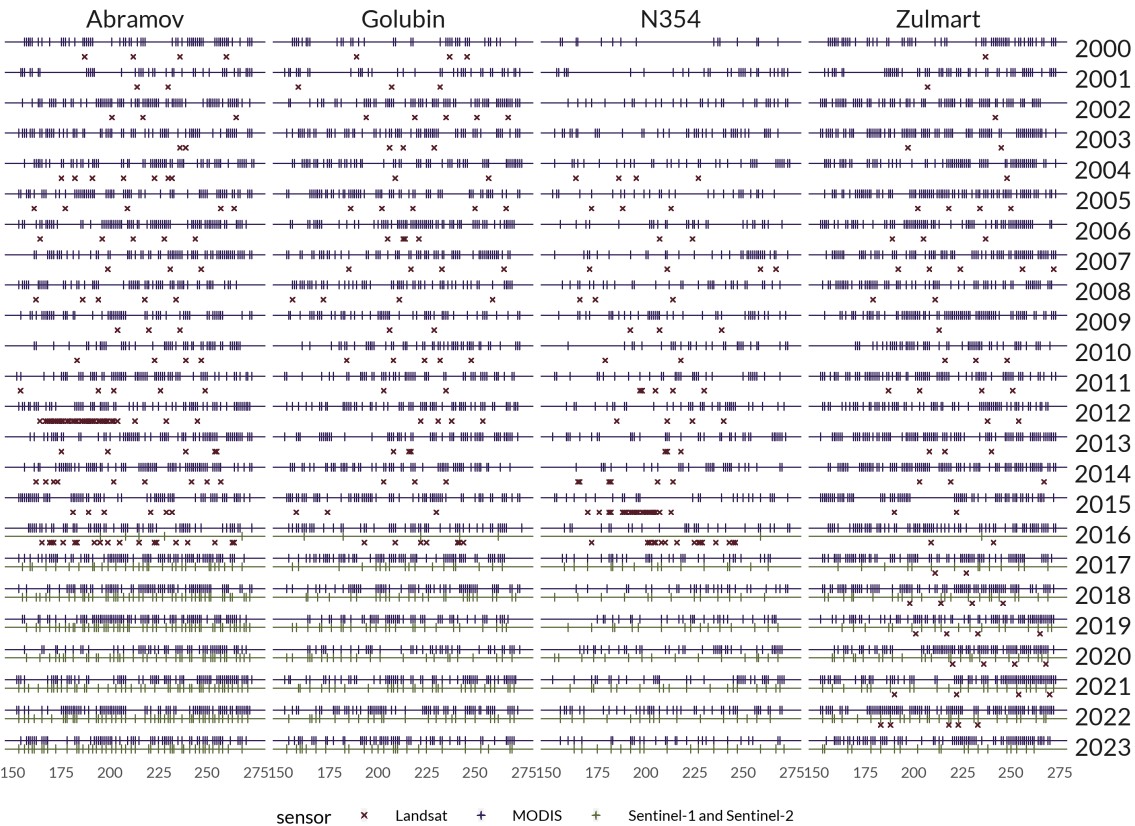

**Figure A1.** Acquisition dates of satellite imagery utilized between 2000 and 2023 for four study glaciers.



## Appendix B: Exponential regression for other glaciers in Central Asia

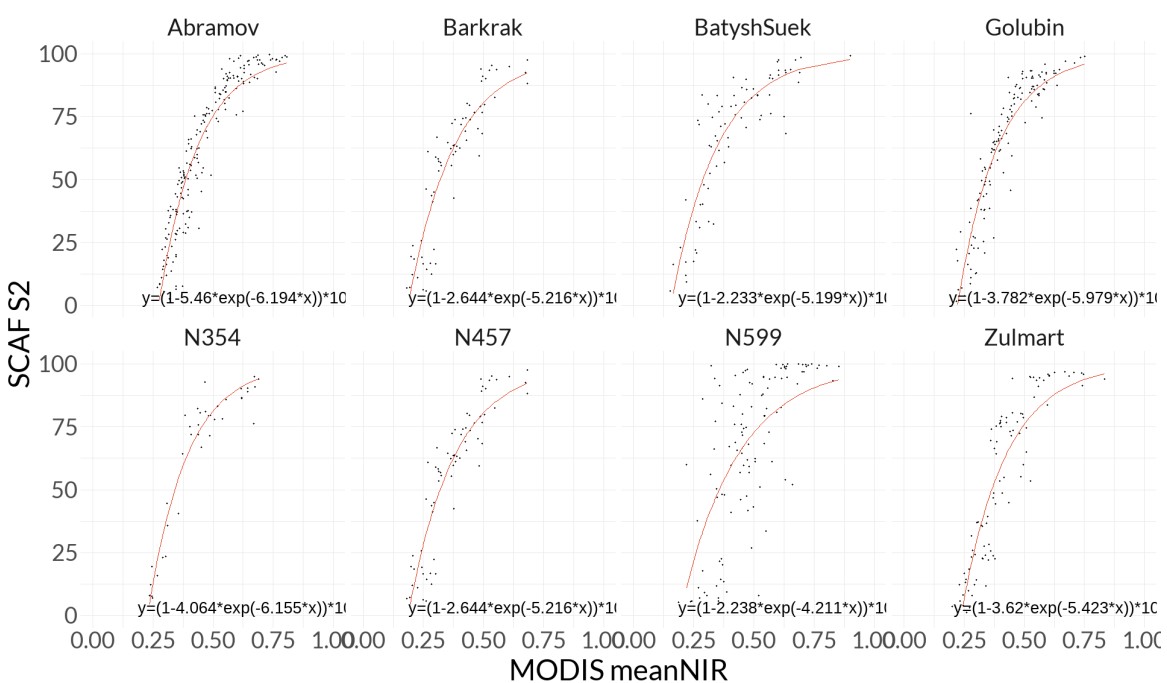

**Figure B1.** Exponential regression derived for other glaciers in Central Asia. Only the SCAF derived from Sentinel-2 used as independent variable. The location of the other glaciers are reported in (Barandun et al., 2025a)



**Appendix C: End-of-season SCAF and annual mass balance correlation**

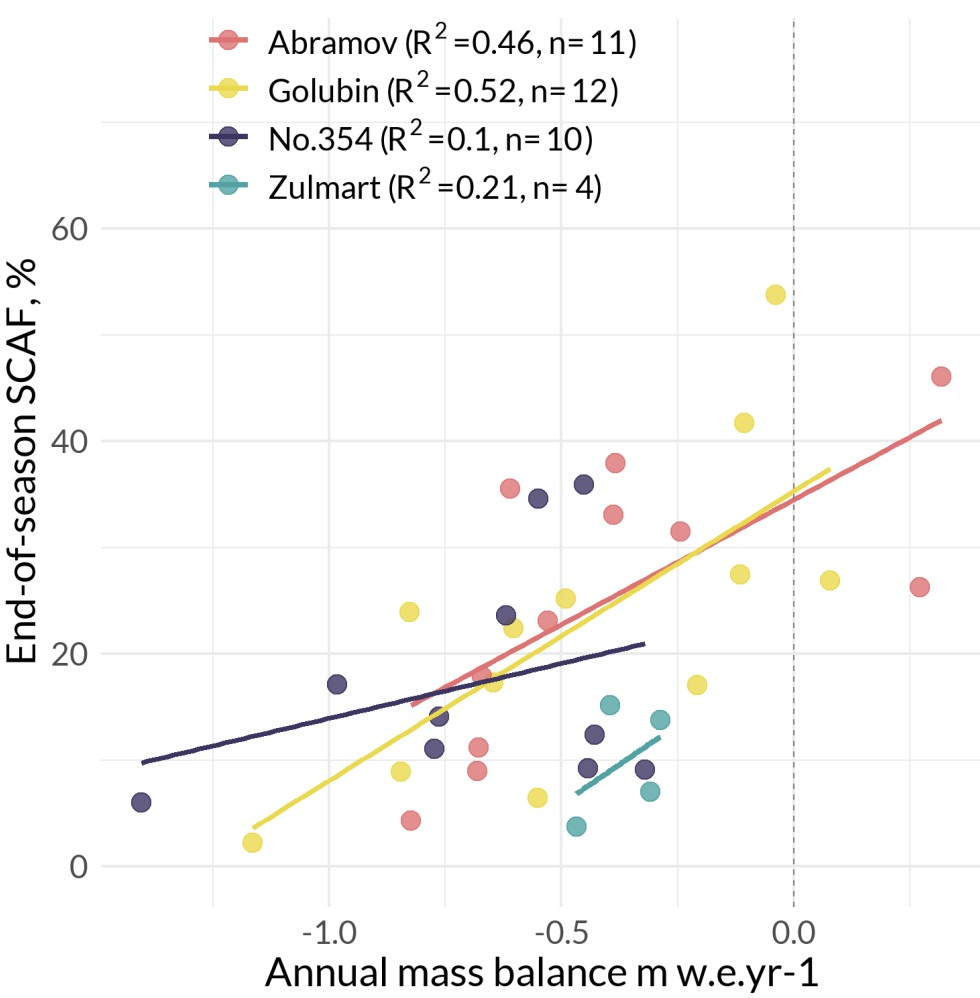

**Figure C1.** Relationship between end-of-season SCAF and annual mass balance for four glaciers in Central Asia. Years where the snowline rose over the glacier area at the end of the season were excluded.

*Author contributions.* MB designed the study. DK developed the methodology with contribution from MC. DK processed, analyzed and contextualized the data. MB, EA and RK provided the measured and modelled SMB time series. DK and MB provided the discussion. All authors contributed to the review of the manuscript.

*Competing interests.* The contact author has declared that neither of the authors has any competing interests.



*Acknowledgements.* This study is supported by Snowline4DailyWater. The project Snowline4DailyWater has received funding from the
Autonomous Province of Bozen/Bolzano – Department for Innovation, Research and University in the frame of the Seal of Excellence Pro-
gramme.We thank the GEF-UNDP-UNESCO funded project "Strengthening the Resilience of Central Asian Countries by Enabling Regional
Cooperation to Assess High Altitude Glacio-nival Systems to Develop Integrated Methods for Sustainable Development and Adaptation to
Climate Change", GEF Project ID 10077. We acknowledge the project "Cryospheric Observation and Modelling for Improved Adaptation in
Central Asia" (CROMO-ADAPT), contract no. 81072443, between the Swiss Agency for Development and Cooperation and the University
of Fribourg, the SPI Flagship Initiative with the project PAMIR (Grant Number: SPI-FLAG-2021-001) and the SPI project Technogrant
(grant no. TEG-2022-001) funded by the Swiss Polar Institute. This study was additionally supported by Horizon Europe - Research and
Innovation - European Union 1011 through the project "Water Efficient Allocation in a Central Asian Transboundary River Basin" 1012
(WE-ACT; contract no. 28000074). We are very grateful to all the research institutes in Central Asia for their input and support in collect-
ing and providing the long-term glacier monitoring data, namely the Central-Asian Institute for Applied Geosciences (CAIAG), the State
Agency for Hydrometeorology under the Ministry of Emergency Situations of the Kyrgyz Republic, Center for Research of Glaciers of the
National Academy of Sciences of the Tajikistan, Agency for Hydrometeorology of the Committee for Environmental Protection under the
Government of the Republic of Tajikistan.





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
