# Peer review of "Sub-seasonal snowline dynamics of glaciers in Central Asia from multi-sensor satellite observations, 2000-2023"

_EGUsphere, 2025_

## Referee Comment (RC2)

The accurate monitoring of sub-seasonal snowline migration is of great necessity since it could provide more detailed SMB dynamics as well as meltwater distribution over glaciers. This manuscript has made nice attempts in this aspect by obtaining close-to-daily SCAF information which could potentially indicate SMB. However, to be honest, I find it a little hard to understand this paper and the language must be further polished. Although the connection between SCAF from Sentinel-1/2 and MODIS NIR exhibits some novelty, the current description about methods seems not so convincing to me, and I have serious doubts about its availability. Unfortunately, I think the current manuscript could not meet the publication standards of this journal.

**Major improvements needed:**

- There are too many errors in citations and punctuation marks which cannot meet the standard of a high-quality paper, please carefully check. If you use LaTex, please carefully check.
- Since the topic is snowline dynamics, it is a bit strange that you did not mention anything about SLA in the Abstract, it is better to add some direct explanations. Another possible way is to change the title to snow-covered area as you demonstrate in Line 47.
- The Introduction section can be streamlined, the current version contains too many unnecessary details, such as the in-situ observations during Soviet times.
- Line 46-47, I find it is still hard to understand due to the intrinsic influences of glacier locations and geometry on its SMB dynamics.
- As for the No.354 Glacier, why did it have the recorded average precipitation lower than the first two glaciers since it experiences frequent summer snowfalls?
- Why didn't the authors employ Sentinel-2 data to manually snowline extraction after 2016? Provide more detailed information about the manual extracted snowline.
- When the authors conducted glacier surface classification, how did you treat other possible landcover types (e.g. rocks) as well as shadowed snow? In addition, only the NIR and backscattering information were utilized, I truly doubt if these properties could comprehensively demonstrate the differences between snow and other types of landcover and the availability of the current SCAF S2 and SCAF S1 methods. How did you determine the Otsu thresholds? Have you confirmed the

availability of these thresholds in regional applications? The current description is confusing and needs modification. Some published studies which rely on SAR information to discriminate glacier zones (e.g. dry snow, wet snow, percolation and bare ice) have failed to generate a continuous snowline in the Antarctic Peninsula. Moreover, there were significant discrepancies even for same area in same year among different studies due to their various thresholds (please see Zhou et al., 2017, Idalino et al., 2024, Arigony-Neto et al., 2020).

- How did the authors treat the different spatial resolution when you merge SCAF S1 and S2 results?
- I don't think there is a nice consistency between MODIS-based SCAF and manual results before the joint employment of S1 and S2 in Figure 4. If possible, please provide some quantitative analysis.
- Line 338, why is this large difference? As for Glacier No.354, the SCAF S1 is close to 0 while that from MODIS-based SCAF is higher than 0.75, which one could represent real conditions? Since there exhibits such significant discrepancies even in 2018, how confident the authors are when fetching SCAF by utilizing the regression functions?
- Line 374, I don't see any trend to a later end of the melt season has become visible since 2009.
- Line 379, what do you mean that snowline rate changed by -1.25%/day? Could you illustrate this by utilizing meters for SLA dynamics? I don't think it is appropriate to directly treat snowline as SCAF.
- It is better to provide a figure with SLA information overlapping with remote sensing images to directly exhibit the identification accuracy.
- Figure 3 to Figure 6, all these figures can be improved.

**Specific comments:**

Line 11, Give some quantitative descriptions about the earlier exposure of bare ice.

Line 13, Considering your research period began from 2000, and some glaciers have already exhibited accelerated snow depletion rates. I wonder how you selected these four glaciers. Is it a bit deliberate?

Line 19, Modify to by the World Meteorological Organization (WMO) in 2022.

Line 22, Add references.

Line 35-40, please add some newly published literature.

Line 57, As for reflectance, give some specific ranges.

Line 60, missing references.

Line 66, which albedo product? Be more specific.

Line 70, how long?

Line 75, give specific number of the coarse resolution.

Line 80, add a brief explanation on the low backscattering of melting snow.

Line 83, missing '.'.

Line 84, which new satellites?

Line 91, correct the misrepresentation. Moreover, 'improve sub-seasonal mass balances and glacier melt water contribution'?

Line 105, missing '.'. There are too many similar problems in the whole manuscript.

Table 1, it should be 'in-situ', and as for locations, missing the specific °N and °E.

Figure 1. missing '.' in the last sentence. Too many similar errors in the manuscript and I won't list each one of them.

Line 107, (?). What do the authors mean?

Line 111, how could it be possible that a 1996 paper includes the data during same period as temperature data collected until 1998?

Line 161, remove 'great'.

Line 166 and 171, add corresponding references.

2.2.2 Radar Data

Line 176, add corresponding references.

Line 233, why used the Italic style?

Line 243, 'land images'?

Line 250, it should be 'similar as in Rastner et al., (2019).'.

Line 256, '(3)'?

Line 276, how did you find the -6 dB is suitable? Need more explanation.

Line 290, why '20%'?

Line 295, Sentinel-1 data failed to detect dry snow? See Zhou et al., 2017, Idalino et al., 2024, Arigony-Neto et al., 2020.

Line 306, why 1.5 SD?

Figure 3, add the corresponding function in each sub-figure.

Line 316, why choose 2000-2023 and 2009-2023? Needs demonstration.

Line 330, where is Fig. 4.1.2??

Line 345, is it a satisfactory accuracy for an RMSE lower than 20%?

Line 391, more negative? Give quantitative description.

Line 439-445, this paragraph has to be re-organized.

Line 461-462, what do you mean?

5.2.2 Uncertainties and limitations, this part also needs to be re-organized.

Line 545-547, this sentence seems unreasonable to me, could the optical based method hamper the utilization of SAR data?

Line 593, could merely four glaciers represent the glacier conditions in the wide Pamir and Tien Shan regions?